# Missing Cases of Bacteriologically Confirmed TB/DR-TB from the National Treatment Registers in West and North Sumatra Provinces, Indonesia

**DOI:** 10.3390/tropicalmed8010031

**Published:** 2023-01-02

**Authors:** Ratno Widoyo, Defriman Djafri, Ade Suzana Eka Putri, Finny Fitry Yani, R Lia Kusumawati, Thakerng Wongsirichot, Virasakdi Chongsuvivatwong

**Affiliations:** 1Epidemiology Unit, Faculty of Medicine, Prince Songkla University, Hat Yai 90110, Thailand; 2Department of Epidemiology and Biostatistics, Faculty of Public Health, Universitas Andalas, Padang 25128, Indonesia; 3Department of Child Health, Faculty of Medicine, Universitas Andalas, Dr. M. Djamil General Hospital, Padang 25128, Indonesia; 4Department of Microbiology, Faculty of Medicine, Universitas Sumatra Utara, H. Adam Malik Hospital, Medan 20136, Indonesia; 5Division of Computational Science, Faculty of Science, Prince of Songkla University, Hat Yai 90110, Thailand

**Keywords:** tuberculosis, missing, cascade analysis, laboratory register, notified cases

## Abstract

This study aimed to assess the percentage of confirmed drug-sensitive (DS) TB and drug-resistant (DR) TB patients who were missing in the national treatment registration in North Sumatra and West Sumatra, where treatment services for DR-TB in North Sumatra are relatively well established compared with West Sumatra, where the system recently started. Confirmed DS/DR-TB records in the laboratory register at 40 government health facilities in 2017 and 2018 were traced to determine whether they were in the treatment register databases. A Jaro–Winkler soundexed string distance analysis enhanced by socio-demographic information matching had sensitivity and specificity over 98% in identifying the same person in the same or different databases. The laboratory data contained 5885 newly diagnosed records of bacteriologically confirmed TB cases. Of the 5885 cases, 1424 of 5353 (26.6%) DS-TB cases and 133 of 532 (25.0%) DR-TB cases were missing in the treatment notification database. The odds of missing treatment for DS-TB was similar for both provinces (AOR = 1.0 (0.9, 1.2), but for DR-TB, North Sumatra had a significantly lower missing odds ratio (AOR = 0.4 (0.2, 0.7). The system must be improved to reduce this missing rate, especially for DR-TB in West Sumatra.

## 1. Introduction

Tuberculosis (TB) is an infectious disease that is a leading cause of human morbidity and death [1]. The case fatality rate of untreated bacteriologically confirmed tuberculosis ranges from 0% to 82% [2]. Globally, 50% of the 40 million targeted tuberculosis cases were treated from 2018 to 2020 [3]. Untreated patients continue to transmit the disease and/or develop drug-resistant (DR) TB [4,5]. A smear-positive patient is able to generate an estimated 10–15 secondary infections in a year [5]. The number of untreated cases has had negative effect on meeting the indicators of the “End TB Strategy“ [6]. To evaluate the factual achievements of the indicators, all countries should have TB as a notifiable disease [7].

Worldwide, under-recognition of the global TB burden resulted from lost to follow-up patients and missing cases [3,7]. Studies in the Middle East and in some African countries reported missing cases ranged from 4% to 38% [8,9,10]. Missing cases are more likely to be more common in a country where the TB referral systems are complicated [11,12]. Health system factors related to missing cases are the need for repeated visits, quality of the recording systems, and delays in receiving the results [8]. Other risk factors for missing cases are living in an urban area, older age, geographical location, diagnosis in a high volume health facility, and diagnosis in a regional hospital [8,13].

Indonesia is a high-TB-burden country that accounts for about 824,000 cases annually [3]. Although the Indonesian National TB Program (NTP) has achieved more than an 80% treatment success rate, the percentage of TB notifications was only 66.5% of estimated cases in 2019 [3]. The government has provided the cartridge-based *Mycobacterium tuberculosis* (MTB)/rifampicin (RIF) assay (Xpert MTB/RIF test) since 2017 [14,15] to scale up the number of TB diagnosed cases. The machine gives a more sensitive result on detecting MTB in sputum in one shot than acid-fast bacilli (AFB) smear microscopy by Ziehl-Neelsen and suitable screening for drug resistance (DR). However, the machine is available only in certain hospitals. Under the new policy, sputum samples or patients must be referred to a center with the Xpert MTB/RIF test. The patients need to wait and return to the health facility.

The costs of the diagnosis and treatment of TB are provided at no charge to the patients as it is covered by the national universal health coverage, known as *Jaminan Kesehatan Nasional/JKN,* and the NTP. Shifting from the AFB smear to the Xpert MTB/RIF should improve the number of diagnosed cases, but a significant proportion of the diagnosed cases might not be notified and treated properly [4,5].

In the provinces of North Sumatra and West Sumatra, where our study was based, the services for DR-TB are more readily available in North Sumatra than in West Sumatra. For culture and drug sensitivity testing, there is only one L3 hospital (Adam Malik Hospital) in Medan in North Sumatra [16]. Furthermore, 17 hospitals of DR-TB treatment centers are available in North Sumatra compared with only 4 hospitals in West Sumatra. The numbers of Xpert MTB/RIF machines were 37 in North Sumatra and 28 in West Sumatra [17]. Although private medical services are available, none in the study area have any Xpert MTB/RIF machines. Diagnosis and treatment of tuberculosis in these two provinces, especially for DR-TB, is nearly exclusively under the governmental health service.

With such a complicated laboratory network and patient referral system, there is a need to evaluate the extent of missing and lost to follow-up patients. The primary objective of this study was to assess the percentages of bacteriologically confirmed DS/DR-TB patients diagnosed in government-owned health facilities that have gone missing from the national TB notification database. The secondary objective was to identify the pitfalls in the health system that led to these missing patients.

## 2. Materials and Methods

### 2.1. Design

This was a cross-sectional study using records from the laboratory registers and national notification databases.

### 2.2. Study Setting

North Sumatra and West Sumatra have populations of 5,259,528 and 14,102,911, respectively. Both provinces have high TB burdens with case notification rates of 150/100,000 in North Sumatra and 149/100,000 in West Sumatra, which ranks them number 5 and number 12, respectively, of 34 provinces [18]. These case notification rates accounted for less than 36% of estimated cases. West Sumatra has 19 districts that include 7 cities and 12 regencies, and North Sumatra has 33 districts that include 8 cities and 25 regencies.

### 2.3. Health System for TB Diagnosis and Treatment

Patient who visit a health center (*L1 Puskesmas*) with signs and symptoms of TB need to have their sputum specimen shipped to an appropriate hospital, or they can go to the hospital on their own for the Xpert MTB/RIF test. Cases that test positive for TB without rifampicin resistance are referred for treatment at the *L1 Puskesmas*. The patients with rifampicin resistance are treated at the hospital. During this transition period many *L1 Puskesmas* still have their own AFB smear test, which means the patients receive the results on the same day and presumptive treatment for drug sensitive TB (DS-TB) can be started. All patients who had either an AFB test or the Xpert MTB/RIF test that showed MTB sensitivity to rifampicin are registered in the local lab databases. Once patients are treated, the cases are recorded in the integrated tuberculosis information system (*Sistem Informasi Tuberculosis Terpadu* [SITT]) for DS-TB and presumptive DS-TB treatments. The rifampicin-resistant cases are treated at referral hospitals as DR-TB patients and the cases are recorded in an electronic tuberculosis management information system (*e-TB Manager).*

### 2.4. Study Population and Sampling Process

Each province has one L3 hospital that provides a TB/DR-TB diagnosis and treatment. These L3 hospitals were included in this study. Since TB management and the reporting system in Indonesia are based on the city and regency, our study chose 4 cities and 5 regencies in West Sumatra and 2 cities and 2 regencies in North Sumatra. The 13 areas were selected based on the distribution of the cities and regencies and the geographical terrains to include coastal, mountainous, and plains areas. In each study area that had only one or two L2 hospitals of government-owned health facilities, we chose one main L2 hospital and two *L1 Puskesmas*. The *L1 Puskesmas* were randomly selected to represent urban and rural areas. Finally, the study included 2 L3 hospitals, 13 L2 hospitals, and 25 *L1 Puskesmas,* but 1 health facility refused to join. The data used in this study were registered as newly diagnosed records of bacteriologically confirmed TB cases from 2017 to 2018 in the local laboratories. The data collection period was from January to August 2019.

### 2.5. Data Source and Flow

Figure 1 summarize data sources and flow. All of the local laboratory data in the study area with an L1 *Puskesmas* and hospital were collected from paper-based and personal-computer (PC)-based systems. The information was entered into EpiData 3.1 software by the first author (RW). Double-entry with validation was performed for data from the paper-based systems. Any discordance of information from the two entries were checked against the paper-based data. The treatment data were extracted directly from the SITT and *e-TB manager* systems with the permission from the relevant authorities.

### 2.6. Analysis of Matching Records

Patients who were excluded from the study were non-residents of the provinces of North and West Sumatra, patients with missing information in the primary fields (i.e., gender, age, district address, or name), and patients younger than 8 years old. After removing non-eligible records, we checked the similarity of records to remove duplicates within the original source data table (paper- and PC-based) and to detect similarities across data tables to link them. Similarities were initially based on socio-demographic criteria. The socio-demographic criteria included gender, age within 2 years, district of residence, and dates within a 6-month difference. With the same group of patients with matched variables, we analyzed further similarities using the Soundex [19] name-matching algorithm with the Jaro–Winkler string distance metric [20,21].

### 2.7. Soundex Analysis

We used the Soundex name-matching methodology for two reasons: (1) to maintain confidentiality of the names and (2) to reduce the problem of different spellings of the same name. The first and second names were combined into one word without a space. This would avoid the problem of unnecessary space characters in the subsequent string distance analysis.

### 2.8. The Jaro–Winkler String Distance

This computational method was used to identify persons with the same or similar ‘soundexed’ names. For each pair of Soundex codes, the Jaro–Winkler score was between 0 (fully matched) and 1 (none matched) [22,23]. Both Soundex and string distance analyses were performed using the ‘stringdist’ package of R software [24] following Equations (1) and (2) of Jaro–Winkler [20].
(1)$$d_{jaro}\left(s,t\right)=1-\frac{1}{3}+\left(\frac{m}{\left|s\right|}+\frac{m}{\left|t\right|}+\frac{m-T}{m}\right),$$
where d*_jaro(s,t)_* is the Jaro distance score, |s| is length of string 1, m is the number of matching characters, |t| is length of string 2, and T is the number of transpositions necessary to turn s’ into t’.
(2)$$d_{jw}\left(s,t\right)=d_{jaro}\left(s,t\right)\left[1-pl\left(s,t\right)\right],$$
where d*_jw(s,t)_* is the Jaro–Winkler distance score, *p* is a user-defined weight (recommended weight *p* = 0.1), and *l*(s,t) is the length of the longest common prefix.

### 2.9. Cut Point and Accuracy of the Jaro–Winkler Distance

Our next task was to identify the appropriate cut point of the Jaro–Winkler score to determine whether two records were actually from the same person. We first obtained a subset of pairs of people in the laboratory dataset and treatment register of the *e-TB Manager*, which had been proven to provide real matches based on citizen ID and socio-demographic variables. For the pairs of different people, we paired subjects based on the above socio-demographic criteria. Since there were too many possible pairs, we randomly chose the same number of unmatched pairs as we did on the matched pairs. The Soundex string distance distribution of these pairs was compared between the proven matched and proven unmatched. The cut point was then identified from the distance value that best discriminated the two data sets. This value was then classified as to whether any other pairs of records in the database were from the same person. In cases of duplication, we kept only the record with the higher accuracy in the MTB diagnosis, i.e., culture with drug sensitivity test > Xpert MTB/RIF > AFB smear.

### 2.10. Linkage Laboratory to Treatment Register

After the duplicated cases were removed, cases listed in the laboratory were cross-checked against those from the treatment register using the same cut point of string distance. Our focus was on the linkage records from the laboratory registers to the treatment registers without emphasis on anti-TB drug patterns. We traced the AFB positive patients and those with DS-TB from the Xpert MTB/RIF in the SITT database. The confirmed DR-TB patients were traced in the *e-TB Manager* database. Since we predefined the specific study period to evaluate cases of missing treatment, patients treated before and after the study period were excluded.

### 2.11. Analysis of Predictors for Missing Cases

All variables were initially analyzed using descriptive statistics such as frequency and percentage. The outcome variable was whether or not the person was missing in a treatment database (SITT for DS-TB or *e-TB Manager* for DR-TB). Independent variables included type of TB, facility levels, provinces, and patient characteristics such as gender, age, address, and year of diagnosis. We used type of TB disease (DS or DR) as the stratified factor because they were reported and referred in different systems as mentioned above. The proportions of missing cases in the subgroups were displayed and computed within the type of TB. The Mantel–Haenszel chi-square test was used initially to check whether the independent variable was associated with the missing patient with adjustment for the type of TB. Furthermore, we carried out a heterogeneity test across the two strata of types of TB. A significant heterogeneity test would suggest that the missing patients in the subgroups of a particular independent variable were significantly different between the DR-TB compared with the DS-TB. The interaction between variables with *p*-values of heterogeneity < 0.05 were included in the multivariate logistic regression. This regression adjusted confounding effects of all independent variables in the model. The effects of independent variables are reported as odds ratios and 95% confidence intervals (CI). For a variable with significant interaction, the odds ratios were reported separately for DS-TB and DR-TB.

### 2.12. Ethical Consideration

Ethical approval was obtained by the Ethics Review Committee of Prince of Songkla University, Thailand (REC: 61-291-18-1), the Committee of the Research Ethics of the Faculty of Medicine, Andalas University (No: 102/KEP/FK/2019), and the Health Research Ethics Committee RSUP Dr. M Djamil Padang (No: 224/KEPK/2019).

## 3. Results

### 3.1. Data Sources

The laboratory records were from 40 out of 942 government-owned facilities (2 out of 4 L3 hospitals, 13 out of 65 L2 hospitals, and 25 out of 871 L1 *Puskesmas*) (Table 1). The number of records was approximately one-eighth of the treatment databases (SITT and *e-TB Manager*). The number of non-residents was higher in the laboratory data than in the treatment database because most non-residents of these two provinces were referred back to their home province for initiation of treatments. The laboratory level revealed a higher frequency of missing information, which reflected the quality of the registration system at the peripheral levels. After removal of duplicate records and application of exclusion criteria, 5,885 records from the laboratory database were checked for linkages in the SITT and *e-TB Manager* databases.

### 3.2. Cut Point for the Soundex String Distance

To analyze a proper cut point for Soundex string similarity, we used only records with a valid citizen ID based on an acceptance range of codes for province, district, and date of birth from the laboratory and *e-TB Manager* databases. Of 2450 pairs matched on socio-demographic variables, 128 were proven to be the same person based on citizen ID and laboratory database. For the remaining 2322 pairs of different citizen ID but the same socio-demographic variables, we randomly chose 128 pairs to be in the unmatched pair control group. Table 2 compares the distribution of the Jaro–Winkler distance of the proven matched and unmatched pair sets. With a distance of 0.134 or less (first two rows), 100% of matched pairs were correctly classified as matched (sensitivity) and 99.2% of unmatched pairs were correctly classified as unmatched (specificity). Therefore, we used this cut point to classify whether any two records were from the same person.

### 3.3. Cascade of Patients and Missing DS-TB/DR-TB Cases

Table 3 shows the distribution of DS- and DR-TB cases that were laboratory diagnosed (in rows) and where they were treated (notified) and missing (in columns). The L1 *Puskesmas* contributed only 1,052 DS-TB diagnoses. The remaining DS-TB and DR-TB cases were diagnosed at L2 and L3 hospitals. This occurred because the majority of the collected sputum specimens were shipped to L2 and L3 hospitals for the Xpert MTB/RIF test. On the other hand, column L1 *Puskesmas* contributed the vast majority of DS-TB treatments (notified). These patients included: (1) DS-TB patients diagnosed and treated in the same L1 *Puskesmas*, (2) patients whose sputum sample test results were returned from the hospital, and (3) patients referred from hospitals to receive DS-TB treatment at the L1 *Puskesmas* near their home.

In the final column of missing patients of Table 3, as expected, those with a laboratory diagnosis made at the L1 *Puskesmas* had the lowest percentage of missing patients (7.2%). The percentages increased as the site of diagnosis of DS-TB climbed up the health service hierarchy. DS-TB diagnosed in an L3 hospital had a missing percentage as high as 40.4%. However, the DR-TB missing pattern was different as the missing percentage was higher in L2 than in L3 hospitals. This occurred because Adam Malik Hospital in North Sumatra had more local DR-TB cases than the combined DR-TB cases from the other study hospitals. The proportion of unlinked or missing patients from notification was 26.6% (95% CI: 25.4–27.8%) for DS-TB including AFB-positive and was 25.0% (95% CI: 21.5–28.8%) for DR-TB.

The first of two columns of Table 4 stratifies the patients into DS-TB and DR-TB. The Mantel–Haenszel chi-square tests in the third column gave non-significant results, which indicated that the variables listed were not associated with missing cases after adjusting for the type of TB. Heterogeneity test *p*-values were >0.05 for variables (except province). These indicated that the effect of these independent variables were similar in both DS-TB and DR-TB. For the province, the heterogeneity *p* value was 0.005. The test suggested that the effect of province in DS-TB and DR-TB was significantly different. As indicated, the missing rates for DS-TB in both provinces were between 26% and 27%. However, for DR-TB the missing rate of 41.4% in West Sumatra was much higher than the 23% missing in North Sumatra.

Since the independent variables could confound one another, we finally used logistic regression to compute their independent effect and displayed the odds ratio and 95% CI in the last column of Table 4. In the logistic regression the interaction for province*type of TB was significant. We then separated the odds ratios of the province North Sumatra and West Sumatra into DS-TB and DR-TB columns. The odds ratio for North Sumatra was 1.0 (0.9,1.2) for DS-TB, indicating no difference in the odds of missing treatment among DS-TB between these provinces. In contrast, the odds ratio for DR-TB in North Sumatra was 0.4 (0.2,0.7), suggesting that the odds of missing treatment in that province are much less than West Sumatra. In summary, there was a significant disparity between these two provinces in DR-TB but not in DS-TB.

## 4. Discussion

Most of the lab tests were registered at the hospitals, except for cases that were referred for treatment in L1 *Puskesmas;* however*,* DR-TB cases were exclusively managed at the hospital. The Jaro–Winkler score of the soundexed names was useful in detecting the same person within and across the databases. This method of checking gave high sensitivity and specificity. Around one-quarter of the notification cases were found to be missing from the notification database. The risk factors for missing treatment included being elderly, being female, being diagnosed in 2018 compared to 2017, and being diagnosed at a hospital. West Sumatra had a similar missing rate as North Sumatra for DS-TB but had a much higher missing rate for DR-TB.

During the study period, TB registration at the laboratory and the notification registry had an inadequate amount of recorded data on personal citizen ID information. This made it difficult to check whether the same person was duplicated in the same database or appeared in both databases. The method we used could solve this problem well. Our findings were similar to previous studies such as those on longitudinal medical records [25,26,27,28] and on linking U.S. historical census data [29]. Therefore, this method may be useful in areas where citizen ID linkage is not ideal.

Most bacteriologically confirmed patients in our setting were diagnosed in L2 hospitals where the Xpert MTB/RIF assay was available. The main method of diagnosis changed from sputum smear microscopy, which is performed at L1 *Puskesmas*, due to a change in policy to promote use of the Xpert MTB/RIF assay [30]. This newer technology increases the sensitivity of a TB diagnosis as well as detecting rifampicin resistant TB at the peripheral laboratories [31]. However, the sputum samples from the non-L2 and non-L3 hospitals had to be shipped to facilities with the Xpert MTB/RIF machines. The shipping increases laboratory turn-around time and requires the patients to make multiple visits before treatment can start [32,33]. This was likely to be the explanation of missing cases in our study.

The percentage of missing DS-TB cases diagnosed at the hospitals were higher than cases diagnosed at the L1 *Puskesmas*. The lost-to-follow-up cases were likely patients who had to wait for shipping of the sputum specimens and return of the Xpert MTB/RIF results from the hospitals. An additional possibility was that the patients were treated in a hospital but notification was not carried out. Currently, the costs of TB treatments are covered by the National Health Insurance Program, known as *Jaminan Kesehatan Nasional/JKN*, for DS-TB, and by the Global Fund for DR-TB [34]. For JKN, reimbursement of the treatment costs to the hospital took place independent of whether the national TB program was notified of the case. This gives no financial motivation for the hospital to complete the notification. This type of pitfall should be corrected by co-regulation of the JKN and the national TB program to improve the reporting system.

Indonesia is well known for under-detection of DR-TB cases. In 2020, 878 Xpert MTB/RIF machines were rolled out to 478 of 514 (93%) districts in Indonesia [34]. The number of confirmed DR-TB cases increased from 2,720 in 2016 to 9,799 cases in 2018 [35]. The DR-TB diagnoses contributed by the Xpert MTB/RIF assay also eliminated time spent on ineffective anti-TB therapy for DR-TB patients. Otherwise, the drug resistance pattern and prevalence would have been extended [36]. However, for the whole community, only 54% of DR-TB patients diagnosed with the Xpert MTB/RIF assay in Indonesia were enrolled for treatment [34]. There is much room for improvement.

A quarter of confirmed cases went missing in this study, which is not small, and it certainly maintains the endemic TB status of the country. Our statistics on lost-to-follow-up patients are higher than the WHO estimates of 10% [3]. Elsewhere, the number of missing DS-TB patients in African studies varied from 6% to 38% [8,9] and in Asia from 4% to 28% [13,37,38]. In DR-TB studies, missing patients were reported to be 15%, 18%, and 37% in West Java in Indonesia [15], Pakistan [39], and South Africa [40], respectively.

We identified several predictors of missing patients in this study. Lost-to-follow-up cases from notifications were significantly greater among the elderly. Older patients were at greater risk of dying while waiting [41], and women were more likely to be missing than men, which may possibly be due to gender inequity [42].

Finally, our data suggested that rapid shifting from the locally available AFB sputum test to a more accurate Xpert test was advantageous. Otherwise, shipping the sputum specimens and patient revisits created the problem of missing treatment. Such transitioning in low- and middle-income countries should therefore be planned carefully to alleviate the problems identified in this study.

The main limitation of the current study was imperfect citizen ID information. The error might bias the result toward increasing missing cases because the same person was not properly registered on both occasions. There is a need to improve the identification system such as embedding the citizen ID cards with an electronic chip that is readable by a card reader. This would eliminate human error from manual data entry.

## 5. Conclusions

A complicated referral system and the shipping of sputum specimens led to TB patients missing their treatments. Therefore, more Xpert MTB/RIF machines should be rolled out to reduce the shipping problems. Identification of the patients using a manual process had poor quality and should be replaced with more advanced chip-based ID cards.

## Figures and Tables

**Figure 1 tropicalmed-08-00031-f001:**
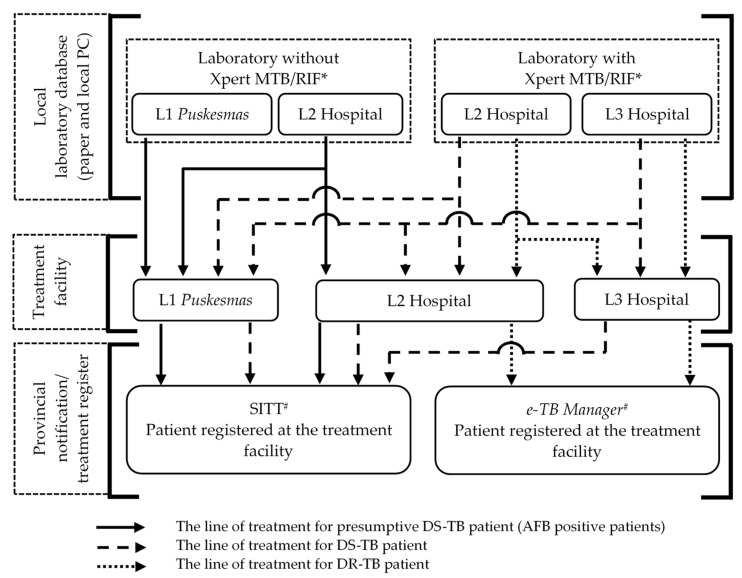
Data sources, referral system, treatment facilities, and provincial treatment register of bacteriologically confirmed TB patients in the provinces of North Sumatra and West Sumatra from 2017 to 2018. DR-TB = drug-resistant tuberculosis, DS-TB = drug-sensitive tuberculosis, *e-TB Manager* = electronic tuberculosis management information system, PC = personal computer, SITT = *Sistem Informasi Tuberculosis Terpadu* (integrated tuberculosis information system). * Laboratory data sources; # notification.

**Table 1 tropicalmed-08-00031-t001:** Number of records identified from each database of North Sumatra Province and West Sumatra Province between 2017 and 2018.

Criteria	Laboratory Register (n)	Provincial Treatment Register	
		SITT (n)	e-TB Manager (n)
Initial records	9353	73,594	794
Removed records	3468	6494	95
Non-residents of North and West Sumatra	543	193	60
Missing information on gender, age, name, or district address	104	67	1
Age less than 8 years	22	3948	-
Duplicated record	1233	2286	34
Previously treated or treatment delayed more than 182 days	1566	-	-
Eligible for linkage analysis	5885	67,100	699
DS-TB	5353	67,100	-
DR-TB	532	-	699

DR-TB = drug resistant tuberculosis; DS-TB = drug sensitive tuberculosis, including AFB-positive patients; e-TB = manager electronic tuberculosis management system; SITT = *Sistem Informasi Tuberculosis Terpadu* (integrated tuberculosis information system).

**Table 2 tropicalmed-08-00031-t002:** Distribution of the Jaro–Winkler distance of matched and unmatched pairs that met the socio-demographic criteria.

Jaro–Winkler Score	Matched (n = 128)	Unmatched (n = 128)
0.000–0.050	126	0
0.051–0.134	2	1
0.135–0.200	0	2
0.201–0.250	0	4
0.251–0.300	0	3
0.301–0.400	0	17
0.401–0.500	0	72
0.501–0.750	0	22
0.751–1.000	0	7

**Table 3 tropicalmed-08-00031-t003:** Cascade of confirmed DS- and DR-TB patients from the diagnosed facility level to the treatment registers in the provinces of North Sumatra and West Sumatra between 2017 and 2018.

Diagnosed Facility Level	Total n	Treatment (Notified) Facility Levels				Missing Patients
		L1 Puskesmas	L2 Hospital	L3 Hospital	Private	
		n (%)	n (%)	n (%)	n (%)	n (%)
DS-TB patients						
All cases	5353	3233 (60.4)	424 (7.9)	91 (1.7)	181 (3.4)	1424 (26.6)
L1 *Puskesmas*	1052	960 (91.2)	9 (0.9)	0 (0.0)	7 (0.7)	76 (7.2)
L2 Hospital	3292	1921 (58.4)	360 (10.9)	6 (0.2)	65 (2.0)	940 (28.6)
L3 Hospital	1009	352 (34.9)	55 (5.5)	85 (8.4)	109 (10.8)	408 (40.4)
DR-TB patients						
All cases	532	0 (0.0)	65 (12.2)	334 (62.8)	0 (0.0)	133 (25.0)
L2 Hospital	132	0 (0.0)	65 (49.2)	5 (3.8)	0 (0.0)	62 (47.0)
L3 Hospital	400	0 (0.0)	0 (0.0)	329 (82.2)	0 (0.0)	71 (17.8)

DR-TB = drug-resistant tuberculosis; DS-TB = drug-sensitive tuberculosis, including AFB-positive patients; L = level.

**Table 4 tropicalmed-08-00031-t004:** Fractions of type of TB patients that were missing in notification database, stratified analysis, and factors associated with missing cases in the provinces of North Sumatra and West Sumatra.

Characteristics	Missing of DS-TB Patients n/Total (%)	Missing of DR-TB Patients n/Total (%)	MH Chi-Square *p* Value	Heterogeneity Test *p* Value	AOR (95% CI) from LR	
Gender						
Female	511/1722 (29.7%)	47/167 (28.1%)	0.434	0.959	**1.3 (1.2, 1.5)**	
Male	913/3631 (25.1%)	86/365 (23.6%)			1	
Age (years)						
8–44	704/2709 (26.0%)	59/264 (22.3%)	0.665	0.364	1	
45–64	502/2032 (24.7%)	64/243 (26.3%)			0.9 (0.8, 1.1)	
65–100	218/612 (35.6%)	10/25 (40.0%)			**1.6 (1.3, 1.9)**	
Patient residence						
Rural	980/3586 (27.3%)	64/264 (24.2%)	0.523	358	1	
Urban	444/1767 (25.1%)	69/268 (25.7%)			1.0 (0.9, 1.1)	
Year of diagnosis						
2017	497/2219 (22.4%)	38/219 (17.4%)	0.416	0.116	1	
2018	927/3134 (29.6%)	95/313 (30.4%)			**1.4 (1.2, 1.6)**	
Facility level						
L1 *Puskesmas*	76/1052 (7.2%)	-			1	
L2 Hospital	940/3292 (28.6%)	62/132 (47%)	not computable		**4.9 (3.8, 6.4)**	
L3 Hospital	408/1009 (40.4%)	71/400 (17.8%)			**7.6 (5.9, 10.0)**	
Province facility level					DS-TB	DR-TB
West Sumatra	801/2977 (26.9%)	24/58 (41.4%)	0.628	0.005 *	1	1
North Sumatra	623/2376 (26.2%)	109/474 (23%)			1.0 (0.9, 1.2)	**0.4 (0.2, 0.7)**

AOR = adjusted odds ratio; CI = confidence interval; DR-TB = drug-resistant tuberculosis; DS-TB = drug-sensitive tuberculosis, including AFB-positive patients; L = level; LR = logistic regression; MH = Mantel–Haenszel. **Boldface** type refers to statistical significance with *p*-values < 0.05.

## Data Availability

Data were obtained from the National TB Program (NTP) and are available in the laboratory register and at the national TB surveillance website (http://sitt.kemkes.go.id/) and the e-TB website (http://etbmanagerbd.org/etbmanager/) with the permission of the Indonesian NTP. The accessed dates were from June, 17 until 24 July 2019.

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
