# Peer review of "Missing Cases of Bacteriologically Confirmed TB/DR-TB from the National Treatment Registers in West and North Sumatra Provinces, Indonesia"

_tropicalmed, 2023, doi:10.3390/tropicalmed8010031_

Round 1

Reviewer 1 Report

Abstract 

Lines 20-23, should be revised especially use of the word 'who went missing'... rather just say .. 'who were missed from the national registry database xxx'

Lines 26- 28 not clear ... eg how many in total were identified in the records, of these how many are active cases, recurrent, use the TB outcomes definitions to guide... then state boldly how many were identified as missing from the records before the proportions and CIs for each category of TB (Susceptible or resistant). This way, it is easier to estimate these values 25.0% (95% CI: 21.5–28.8%) or consider providing actual numbers. 

Last sentence should be a recommendation/conclusion statement. The present sentence should be revised according as it still seems to imply factors associated with missing cases from the cascade. In addition, this last sentence is similar to prior sentence on factors associated.. which need revision as well. As example, instead of female, say gender... again, diagnosed in 2018? relevance? consider revising 

Methods

Study design - this study is more of a cross sectional study involving the use of a secondary data than cohort study. Authors should consider revising this... 

Sub title 'study setting'- reads more like study setting and sampling... for sampling, provide details on specific sampling technique used to identify the study districts and clinics. Also provide information on why North Sumatra and West Sumatra were chosen, How clinically significant is this location in Indonesia in terms of the findings' generalizability? There was just a fair attempt on this in the introduction section lines line 76-77.

Data analysis - mention specific statistics like descriptive and inferential before the significance level considered.

Discussion

The percentage of DR-TB 320 missing cases was higher than DS-TB cases in West Sumatra; however, the opposite was 321 true in North Sumatra.... discuss the clinical implications of this for the province and county... especial for stake holders...

Lines 326-331, consider revising because there is too much emphasis on this 'Jaro-Winkler' besides it seems more like a repetition of what you provided in the first paragraph...

Lines 330 -331, consider moving to limitation section.

Conclusion - recommend more practical strategies that the Indonesian government need to implement to improve rates of missing cases based on the study findings.

Line 210, delete the word '...duplications removed' it has been mentioned already in the data management section

Author Response

Dear Reviewer

Thank you very much for your important review. The authors appreciate your valuable suggestions, which will improve the quality of this manuscript. All editing was made in the main text of the manuscript with marked up using the “Track Changes” function. The questions raised by you were revised as follows and pointed with page and line numbers (all Markup).

Best Regards 

Reviewer 2 Report

The authors have tried to assess the pre-treatment lost to follow-up (PTLFU) and associated factors among bacteriologically confirmed TB patients in Indonesia during 2017-2018. This is a relevant operational research providing fair information to the National TB programme on the extent of PTLFU and the potential vulnerable groups with high PTLFU. In turn, this information is crucial for adjusting the TB burden estimates in the country. In this regard, I congratulate the authors on this effort. However, I have some concerns with the manuscript in the current form and I feel there is scope for improving the manuscript. Please find below the specific comments

Introduction:

1. I feel the introduction is not flowing well. Initially, in the first paragraph you introduce the global burden of 'missing cases' and immediately jump into burden in Indonesia. later, you again discuss about the global context. Instead, you can consider structuring it as below

- Paragraph-1: Global burden of TB and Global commitments to end TB by 2030 (End TB strategy- Pillar 1) 

- Paragraph-2: Missing cases (including undetected and detected but not treated), reasons for missing cases-especially pre-treatment lost to follow-up (diagnosed but not initiated on treatment), global estimates and factors associated with PTLFU from other settings

- Paragraph-3: TB burden in Indonesia, missing TB cases, anecdotal evidence on PTLFU and potential programmatic reasons for PTLFU (here you can briefly highlight the complicated sample referral system in the country- detailed description can be presented in the methods section) 

- Paragraph 4: Rational for the study, novelty of the study and study objectives.

2. Please consider moving the description of study setting (level of health facilities, referal mechanism etc into methods as a  detailing of Figure-1 in the study setting

3. Relook into the references you have used and the way it has been cited. Especially reference 2, 4 and 5. You can consider using information from Global Tuberculosis Report 2022 instead of 2021 and 2020

Methods:

1. Mention what sampling method was used to select the districts. Also, how L2 and L1 facilities were selected for the study.

2. Line 144: What version of EpiData was used? EpiData Manager or EpiData classic? Whether double entered data was validated? If validated, please mention it.

3. Line 161-163: The 'MTB accuracy' is not clear. Please describe the process you adopted. 

4. Line 211-213: Was it a predictive model you were interested in or the explanatory model? What regression method was used? Kindly specify how the variables were included in the final model

5. Who extracted the data from laboratory register? What was their qualification? Howe were they trained? 

6. What percentage discrepancy you found in the 'name' field  when you validated the double entered data in the EpiData. This can mentioned as a measure of quality in data entry (especially names which are the key link variables)

7. Is there any specific reason why the retreatment cases were left out?

Results:

1. In Table-3, provide the additional rows to provide the notification and missing cases for all DS-TB and DR-TB. Thus, you can safely remove Figure-2.

2. Table-4 and Table-5 can be combined together. I am not able to appreciate the utility of chi-square and heterogeneity test here. Instead you can take DS and DR-TB as one the explanatory variable. So that you can have a single table with Explanatory (Independent) variable, Total, Missing cases (%), unadjusted measure of association (95% CI) and adjusted measure of association (95% CI)

3. I strongly recommend use of relative risk over odds ratio. Immediate reason is that the study is a cohort design. Also, when your outcome of interest is >10% (missing cases is 26%), the odds ratio overestimates the association and also the confidence intervals are not precise.

Discussion:

1. Line 314: 'Most of the cases were bacteriologically confirmed', where does this information comes from? I do not see this in the results. 

2. Line 331: Any specific reason why this may not be useful in any place other than the study setting? It had a good analytical validity. Do you think any external factor can harm the analytical validity in other settings? Or you just trying to be cautious in extrapolating the results? Need more thoughts on this. 

3. Line 347-353: Very interesting information. In this context why you should not try JKN database to see what percentage of the 'missing cases' had actually had received the payment for TB care. This would make your estimate more valid. Please ignore this comment if it is too late to do it. However, if this is not addressed, please mention this as a main limitation and would have led to overestimation of missing cases. Or use 'not notified' instead of 'missing cases'

4. There is too much discussion on Xpert/MTB Rif assay, I really do not see the utility of it

5. Please highlight the implications of the study and also related recommendations, may be by taking insights from other settings on the interventions which have worked to reduce PTLFU, especially when diagnosed in secondary or tertiary settings. 

Author Response

(The authors gave the same response as above.)

Round 2

Reviewer 1 Report

All comments were addressed. Thank you